# Anchor-Changing Regularized Natural Policy Gradient for Multi-Objective Reinforcement Learning

**Ruida Zhou**[*]
Texas A&M University
ruida@tamu.edu

**Tao Liu**[*]
Texas A&M University
tliu@tamu.edu

**Dileep Kalathil**
Texas A&M University
dileep.kalathil@tamu.edu

**P. R. Kumar**
Texas A&M University
prk@tamu.edu

**Chao Tian**
Texas A&M University
chao.tian@tamu.edu

## Abstract

We study policy optimization for Markov decision processes (MDPs) with multiple reward value functions, which are to be jointly optimized according to given criteria such as proportional fairness (smooth concave scalarization), hard constraints (constrained MDP), and max-min trade-off. We propose an Anchor-changing Regularized Natural Policy Gradient (ARNPG) framework, which can systematically incorporate ideas from well-performing first-order methods into the design of policy optimization algorithms for multi-objective MDP problems. Theoretically, the designed algorithms based on the ARNPG framework achieve $\tilde{O}(1/T)$ global convergence with exact gradients. Empirically, the ARNPG-guided algorithms also demonstrate superior performance compared to some existing policy gradient-based approaches in both exact gradients and sample-based scenarios.

## 1 Introduction

In many sequential decision-making scenarios, agents usually face multiple objectives simultaneously. This motivates the study of reinforcement learning (RL) with multiple reward values $V_{1:m}^\pi(\rho)$.[2] Given the achievable region $\mathcal{V} = \{V_{1:m}^\pi(\rho)\}_{\pi \in \Pi}$ consisting of value vectors achieved by policies in policy class $\Pi$, the agent employs certain criteria to reflect the system requirement. For example,

1. Proportional fairness [13]: Given $a_{1:m} > 0$, find $v \in \mathcal{V}$ that $\sum_{i=1}^m a_i \frac{v_i' - v_i}{v_i} \leq 0$, $\forall v' \in \mathcal{V}$.
2. Hard constraints [4]: Given $b_{2:m}$, maximize$_{v \in \mathcal{V}}$ $v_1$, subject to $v_i \geq b_i, \forall i = 2, \ldots, m$.
3. Max-min trade-off [8]: Given $c_{1:m} > 0$, maximize$_{v \in \mathcal{V}}$ min$_{i \in [m]}$ $(v_i/c_i)$.

We study policy gradient-based approaches that optimize over parameterized policies $\Pi = \{\pi_\theta : \theta \in \Theta\}$ through policy gradient. In general, the optimization problems above may not be convex in terms of $\theta$, not even for single-objective MDPs with direct parameterization by $\theta_{s,a} = \pi_\theta(a|s)$ [2]. Due to the non-convexity, $O(1/T)$ global convergence of policy gradient-based methods was only established very recently for single-objective MDPs with exact gradients [2, 20]. These breakthrough results have motivated the study of policy optimization for multi-objective MDPs, e.g., smooth concave scalarization [5], constrained MDPs (CMDPs) [11, 31].

However, under the exact gradients scenario, the previous approaches for multi-objective MDPs, either suffer from slow provable $O(1/\sqrt{T})$ global convergence [11], or require extra assumptions

---

[*]The first two authors contributed equally.

[2]The notations are formally defined in Section 2.

36th Conference on Neural Information Processing Systems (NeurIPS 2022).

[37, 33, 18]. The compactness of $\Theta$ is assumed in [37], but this assumption forbids a very common softmax parameterization, where $\Theta = \mathbb{R}^{|\mathcal{S}||\mathcal{A}|}$. The NPG-based methods have been analyzed in [33, 18] under an ergodicity assumption, but such an assumption is not required for NPG in single-objective MDPs [2], and therefore appears artificial.

The above criteria for multi-objective MDPs could be viewed as convex optimization problems w.r.t. a value vector $v \in \mathcal{V}$, for which there are a wide array of well-performing first-order methods for convex optimization problems in general. It is desirable to take full advantage of such efficient first-order methods in a unified and flexible manner when designing policy gradient-based algorithms for multi-objective MDPs.

**Main contributions**

1. We propose an anchor-changing regularized natural policy gradient (ARNPG) framework in Section 3 that can exploit and integrate first-order methods for the design of policy gradient-based algorithms for multi-objective MDPs.
2. We demonstrate the strength of the ARNPG framework by designing algorithms for three general criteria: smooth concave scalarization (Section 4.1), constrained MDPs (Section 4.2), and max-min trade-off (Section 4.3).
3. Under softmax parameterization with exact gradients, the proposed algorithms inherit the advantages of the integrated first-order methods, and are guaranteed to have $\tilde{O}(1/T)$ global convergence without further assumptions on the underlying MDP.
4. In addition to the theoretical advantages, we provide the results of extensive experimentation in Section 5 and Appendices A and B which demonstrate that the ARNPG-guided algorithms provide superior performance in exact gradient and sample-based tabular scenarios, as well as actor-critic deep RL scenarios, compared to several existing policy gradient-based approaches.

## 1.1 Related works

Policy gradient (PG)-based methods have drawn much attention recently [1, 20, 10, 14] due to their simplicity as well as the potential to generalize to large scale problems. Despite their non-convex nature, PG-based methods have been shown to converge globally for single-objective MDPs [1, 20]. Their convergence may be further accelerated with appropriate regularization [10, 17], e.g., entropy regularization, but the algorithms only converge to the optimum of the regularized problem instead of the desired (unregularized) problem.

This paper considers *single-policy* multi-objective MDPs, including CMDPs where constraints are specified on some objectives. Global convergence of PG-based approaches in the multi-objective MDPs has been previously studied. For smooth concave scalarization, Bai et al. [5] showed an $O(1/\epsilon^4)$ sample complexity (to achieve $\epsilon$-optimal in expectation) of the policy-gradient method under sample-based scenarios. However, with exact gradients, we are unaware of works with fast $\tilde{O}(1/T)$ convergence. For CMDPs, Ding et al. [11] have studied a primal-dual NPG algorithm achieving $O(1/\sqrt{T})$ global convergence for both the optimality gap and the constraint violation. Xu et al. [31] have proposed a primal approach that reduces constraint violations with a higher priority than optimizing objective, and enjoys the same $O(1/\sqrt{T})$ global convergence. In work conducted concurrently with ours, [33] and [18] have proposed algorithms that achieve $\tilde{O}(1/T)$ convergence but with extra ergodicity assumptions.

A general setting of optimizing a concave function of the state-action visitation distribution has been considered in [37]. Though the problem is more general, its gradient estimation is more complicated than the canonical policy gradient estimate. Zhang et al. [37] showed that the gradient ascent achieves $O(1/T)$ global convergence for smooth scalarization with exact gradients, under several assumptions such as convexity and compactness of the parameter set $\Theta$. Directly viewing the state-action visitation as the decision variables and imposing equality constraints for their feasibility, a smooth concave scalarization has been studied in [36] and later generalized to the constrained setting in [6]. These two works focus on sample-based scenarios, but due to their primal-dual approach with equality constraints, the convergence rate is only $O(1/\sqrt{T})$ even with exact gradients. Moreover, the state-action visitation parameterization is difficult to generalize to larger scale deep RL scenarios.

A more thorough discussion on related works is given in Appendix F.

## 2 Preliminaries

**System model** A Markov decision process (MDP) is represented by a tuple $(\mathcal{S}, \mathcal{A}, P, \rho, \gamma, r)$, where $\mathcal{S}$ is the state space, $\mathcal{A}$ the action space, $P : \mathcal{S} \times \mathcal{A} \to \Delta(\mathcal{S})$ the transition kernel, $\rho \in \Delta(\mathcal{S})$ the initial state distribution, $\gamma \in (0, 1)$ the discount factor, and $r : \mathcal{S} \times \mathcal{A} \to [0, 1]$ the reward function. Given any policy $\pi : \mathcal{S} \to \Delta(\mathcal{A})$ and any reward function $r : \mathcal{S} \times \mathcal{A} \to [0, 1]$, we define the state value function $V_r^\pi : \mathcal{S} \to [0, \frac{1}{1-\gamma}]$, and the state-action value function $Q_r^\pi : \mathcal{S} \times \mathcal{A} \to [0, \frac{1}{1-\gamma}]$, as

$$V_r^\pi(s) := \mathbb{E}[\sum_{t=0}^\infty \gamma^t r(s_t, a_t) \mid s_0 = s, \pi], \quad Q_r^\pi(s, a) := \mathbb{E}[\sum_{t=0}^\infty \gamma^t r(s_t, a_t) \mid s_0 = s, a_0 = a, \pi],$$

where expectation $\mathbb{E}$ is taken over the random trajectory of the Markov chain induced by the policy $\pi$ and the transition kernel $P$. With a slight abuse of notation, we denote $V_r^\pi(\rho) := \mathbb{E}_{s \sim \rho}[V_r^\pi(s)]$. Define the discounted state-action visitation distribution (state-action visitation for short) of policy $\pi$ with initial state distribution $\rho$ by $d_\rho^\pi(s, a) := (1 - \gamma)\mathbb{E}_{s_0 \sim \rho}[\sum_{t=0}^\infty \gamma^t \mathbb{P}(s_t = s, a_t = a | s_0, \pi)]$. It then follows that $V_r^\pi(\rho) = \frac{1}{1-\gamma}\langle d_\rho^\pi, r \rangle$ by viewing $d_\rho^\pi$ and $r$ as $|\mathcal{S}||\mathcal{A}|$-dimensional vectors indexed by $(s, a) \in \mathcal{S} \times \mathcal{A}$. When it is clear from the context, we denote the state visitation distribution by $d_\rho^\pi(s) := \mathbb{E}_{s_0 \sim \rho}[(1 - \gamma)\sum_{t=0}^\infty \gamma^t \mathbb{P}(s_t = s | s_0)]$, which is the marginal distribution of the state-action visitation $d_\rho^\pi(s, a)$, i.e., $d_\rho^\pi(s) = \sum_{a \in \mathcal{A}} d_\rho^\pi(s, a)$.

We study an MDP with $m$ objectives represented by $(\mathcal{S}, \mathcal{A}, P, \rho, \gamma, r_{1:m})$, where $r_i : \mathcal{S} \times \mathcal{A} \to [0, 1]$ is the $i$-th reward function for each $i \in [m]$. For simplicity, denote $V_i^\pi(\cdot) := V_{r_i}^\pi(\cdot)$ and $V_{1:m}^\pi(\cdot) := (V_1^\pi(\cdot), \ldots, V_m^\pi(\cdot))$. We consider parameterized policies in $\Pi = \{\pi_\theta : \theta \in \Theta\}$, where $\Theta \subset \mathbb{R}^n$ is the parameter space. For example, the softmax policy is $\pi_\theta(a|s) = \frac{\exp(\theta_{s,a})}{\sum_{a'} \exp(\theta_{s,a'})}$ with $\Theta = \mathbb{R}^{|\mathcal{S}||\mathcal{A}|}$; and neural softmax policy is $\pi_\theta(a|s) = \frac{\exp(\text{NN}_\theta(s,a))}{\sum_{a'} \exp(\text{NN}_\theta(s,a'))}$, where $\text{NN}_\theta$ is some neural network parameterized $\theta$. Define $\mathcal{V} := \{V_{1:m}^{\pi_\theta}(\rho) : \theta \in \Theta\}$ as the achievable region of value vectors. The agent wishes to optimize the policy in $\Pi$ for a given specific multi-objective criterion on value vectors in $\mathcal{V}$.

**Mirror ascent** As one of the most well-known iterative optimization methods, mirror descent (actually ascent in the context of our formulation as a maximization problem) [21, 7] is a general class that encompasses many first-order methods in convex optimization. Given a variable $x$ in a compact convex set $\mathcal{X} \subset \mathbb{R}^n$ and an ascent direction $g \in \mathbb{R}^n$, the variational representation of the mirror ascent update is

$$x' \in \arg\max_{y \in \mathcal{X}}\{\langle g, y \rangle - \alpha B_h(y||x)\}, \tag{1}$$

where $B_h(x||y) := h(x) - h(y) - \langle \nabla h(y), x - y \rangle$ is some Bregman divergence generated by a differentiable convex function $h : \mathcal{X} \to \mathbb{R}$. When analyzing the convergence of first-order methods, certain fundamental inequalities are usually established to facilitate the proof. One such inequality is

$$\langle g, x' \rangle - \alpha B_h(x'||x) \geq \langle g, y \rangle - \alpha B_h(y||x) + \alpha B_h(y||x'), \quad \forall y \in \mathcal{X}, \tag{2}$$

which is a critical step in many previous works, e.g., [22, 27, 16].

It is desirable to construct a similar fundamental inequality for multi-objective MDPs that can facilitate the analysis of convergence. As we will show in the next section, such an inequality can indeed be established in a new framework, which we refer to as the Anchor-Changing Regularized Natural Policy Gradient (ARNPG).

**Notations** Denote KL-divergence between two $n$-dimensional probability vectors $x, y$ by $D(x||y) := \sum_{i=1}^n x_i \log(x_i/y_i)$, which is a widely-used Bregman divergence. For any policies $\pi, \pi'$ and state visitation distribution $d$, define $D_d(\pi||\pi') := \sum_{s \in \mathcal{S}} d(s)D(\pi(\cdot|s)||\pi'(\cdot|s))$. A *uniform policy* is one which chooses actions uniformly at random.

## 3 Anchor-changing regularized natural policy gradient

Let us consider a hypothetical mirror ascent update on decision value vector $v_k \in \mathcal{V}$ according to (1). Given an ascent direction $\tilde{G}_k$ along which to improve $v_k$, the updated value vector is

$$v' \in \arg\max_{v \in \mathcal{V}}\{\langle \tilde{G}_k, v \rangle - \alpha B_h(v||v_k)\}. \tag{3}$$

Suppose the value vector $v_k$ is achieved by a policy $\pi_{\theta_k}$, i.e., $v_k = V_{1:m}^{\pi_{\theta_k}}(\rho)$. Denote the reward function in the ascent direction as $\tilde{r}_k(s, a) = \langle \tilde{G}_k, r_{1:m}(s, a) \rangle$. It follows that $\langle \tilde{G}_k, v_k \rangle = V_{\tilde{r}_k}^{\pi_{\theta_k}}(\rho)$. Note that $B_h(v \| v_k)$ in (3) serves the role of a soft constraint on $v$ by keeping $v$ within a vicinity of $v_k$. Replacing $B(v \| v_k)$ by $\frac{D_{d_\rho^{\pi_\theta}}(\pi_\theta \| \pi_{\theta_k})}{1 - \gamma}$ will induce a similar soft constraint that prefers the vicinity of the "anchor" policy $\pi_{\theta_k}$. Therefore we consider replacing the variational update in (3) by

$$\theta' \in \arg\max_{\theta \in \Theta} \left\{ \tilde{V}_{k,\alpha}^{\pi_\theta}(\rho) \right\}, \quad \text{where} \quad \tilde{V}_{k,\alpha}^{\pi_\theta}(\rho) := V_{\tilde{r}_k}^{\pi_\theta}(\rho) - \alpha \frac{D_{d_\rho^{\pi_\theta}}(\pi_\theta \| \pi_{\theta_k})}{1 - \gamma}. \quad (4)$$

**ARNPG** Motivated by the intuition above, we propose the Anchor-Changing Regularized Natural Policy Gradient (ARNPG) framework. At (macro) step $k$, the ARNPG framework determines the reward function in the ascent direction $\tilde{r}_k$ and the anchor policy $\pi_{\theta_k}$, which can exploit well-performed first-order methods in convex optimization literature utilizing the features of the specific criteria in use. With $\tilde{r}_k$ and $\pi_{\theta_k}$, we wish to solve for (4) to improve the value vector. However the optimal solution $\theta'$ of (4) is generally not determinable explicitly. ARNPG therefore approaches the optimal solution via a subroutine that executes a natural policy gradient (NPG) algorithm w.r.t. the KL-regularized value function $\tilde{V}_{k,\alpha}^{\pi_\theta}(\rho)$. We refer to this subroutine, given in Algorithm 1, as InnerLoop($\tilde{r}_k, \pi_{\theta_k}, \alpha, \eta, t_k$). It iteratively updates the parameter $\theta_k^{(t)}$ for $t_k$ (micro) steps according to the NPG update rule as in (5), where $\mathcal{F}_\rho(\theta)^\dagger$ is the Moore-Penrose inverse of the Fisher information matrix $\mathcal{F}_\rho(\theta) := \mathbb{E}_{(s,a) \sim d_\rho^{\pi_\theta}} \left[ \nabla_\theta \log \pi_\theta(a|s) \left( \nabla_\theta \log \pi_\theta(a|s) \right)^\top \right]$.

---

**Algorithm 1:** InnerLoop($\tilde{r}_k, \pi_{\theta_k}, \alpha, \eta, t_k$)

---

**Initialize** $\theta_k^{(0)} = \theta_k$
**for** $t = 0, 1, \ldots t_k - 1$ **do**
$$\quad \theta_k^{(t+1)} \leftarrow \theta_k^{(t)} + \eta \mathcal{F}_\rho(\theta_k^{(t)})^\dagger \nabla_\theta \tilde{V}_{k,\alpha}^{\pi_k^{(t)}}(\rho) \quad (5)$$
**Return** $\theta_k^{(t_k)}$

---

The choice of the number of iterations in InnerLoop (i.e., $t_k$) involves a trade-off between the variational update precision and the overall efficiency. On the one hand, a larger $t_k$ leads to a more accurate approximation of the optimal solution $\theta'$ to (4), but it may cause the algorithm to spend unnecessary computational resources on the regularized objective $\tilde{V}_{k,\alpha}^{\pi_\theta}(\rho)$, instead of on the true optimization problem. On the other hand, a smaller $t_k$ saves inner loop iterations but the update follows less closely to the underlying mirror-ascent update in improving the value vector. In our experiments, we choose $t_k$ within 10 to strike a balance and empirically observe $t_k > 1$ has better performance.

We note that when $t_k = 1$, the gradient $\nabla_\theta \tilde{V}_{k,\alpha}^{\pi_{\theta_k}}(\rho) = \nabla_\theta V_{\tilde{r}_k}^{\pi_{\theta_k}}(\rho)$, since $D_{d_\rho^{\pi_\theta}}(\pi_\theta \| \pi_{\theta_k})$ has zero gradient at $\theta = \theta_k$. The update in (5) reduces to an NPG update on the unregularized value function $\tilde{V}_{\tilde{r}_k}^{\pi_\theta}(\rho)$. For single-objective MDPs, it reduces to the canonical NPG method.

### 3.1 Theoretical guarantee of ARNPG

We now present the main theoretical tool for the analysis of the ARNPG framework. Recall the discussion of the fundamental inequality after (2). Proposition 1 establishes such a fundamental inequality with controllable approximation error under the softmax policy parameterization, i.e., $\pi_\theta(a|s) = \frac{\exp(\theta_{s,a})}{\sum_{a'} \exp(\theta_{s,a'})}$. In the rest of the paper, we omit $\theta$ in $\pi_\theta$ when it is clear from the context, but it should be noted that all updates of policies are performed on the parameters.

**Proposition 1.** *Under the softmax parameterization, given $\epsilon_k > 0$, for any $\tilde{r}_k$, $t_k \geq \frac{1}{1-\gamma} \log(\frac{5\|\tilde{r}_k\|_\infty}{(1-\gamma)^2 \epsilon_k}) + 1$, $\alpha > 0$ and $\eta = \frac{1-\gamma}{\alpha}$, the update $\pi_{k+1} \leftarrow$ InnerLoop$(\pi_k, \tilde{r}_k, \alpha, \eta, t_k)$ satisfies*

$$V_{\tilde{r}_k}^{\pi_{k+1}}(\rho) - \alpha \frac{D_{d_\rho^{\pi_{k+1}}}(\pi_{k+1} \| \pi_k)}{1 - \gamma} \geq V_{\tilde{r}_k}^{\pi}(\rho) - \alpha \frac{D_{d_\rho^\pi}(\pi \| \pi_k) - D_{d_\rho^\pi}(\pi \| \pi_{k+1})}{1 - \gamma} - \epsilon_k, \quad \forall \pi. \quad (6)$$

The inequality (6) is critical to the convergence proof. Its right hand side allows telescoping, which by summing over $k$ can iteratively cancel the terms $D_{d_\rho^\pi}(\pi||\pi_k)$. Since $t_k = \Theta(\log(1/\epsilon_k))$ it suffices to use very few iterations in InnerLoop for maintaining precision.

*Remark.* It has been shown that for the entropy-regularized MDP, i.e., KL-regularized with the uniform policy as the anchor policy, NPG converges linearly (i.e., geometrically fast) to the regularized optimal policy [10]. It is natural to anticipate that for the KL-regularized MDP $\tilde{V}_{k,\alpha}^\pi(\rho)$ with anchor $\pi_k$, NPG would similarly converge linearly (i.e., $\tilde{V}_{k,\alpha}^{\pi_k} \geq \tilde{V}_{k,\alpha}^{\pi_k^*} - \epsilon$ for $t_k = \Theta(\log(1/\epsilon))$) to a corresponding optimal policy, denoted as $\pi_k^*$. In contrast, the right hand side of inequality (6) has a *positive drift* $\alpha \frac{D_{d_\rho^\pi}(\pi||\pi_{k+1})}{1-\gamma}$ *for any policy* $\pi$, which is considerably stronger.

*Proof sketch of Proposition 1.* We can show that InnerLoop approximately solves the variational update in (4) with linear convergence as anticipated. However to establish (6), the difficulty lies in the introduction of positive drift, since $V_{\tilde{r}_k}^{\pi_\theta}(\rho)$ is not concave w.r.t. $\theta$ and $D_{d_\rho^{\pi_\theta}}(\pi_\theta||\pi_{\theta_k})$ may not be a Bregman divergence. We tackle this difficulty by showing that optimizing $\pi_\theta$ in InnerLoop implicitly performs a mirror ascent update for state action visitation $d_\rho^{\pi_\theta}$. □

As demonstrated in the next section, Proposition 1 ensures that the convergence rate of the algorithms derived from the ARNPG framework is of the same rate as the underlying first-order methods with only extra logarithmic factors.

## 4 Theoretical applications

In this section, we apply the ARNPG framework to several important multi-objective MDP scenarios and obtain new policy optimization algorithms by integrating first-order methods in convex optimization. All the theoretical results presented in this section are under the softmax parameterization with exact gradients. However, the obtained algorithms can be implemented in more general settings such as neural softmax and sample-based scenarios, as in the next section. We theoretically establish $\tilde{O}(1/T)$ convergence of these algorithms by leveraging the fundamental inequality in Proposition 1.

### 4.1 Smooth concave scalarization function

We start by considering the following optimization problem
$$\max_\theta F(V_{1:m}^{\pi_\theta}(\rho)), \tag{7}$$
where $F$ is a concave function, and $\beta$-smooth w.r.t. $\|\cdot\|_\infty$ norm, i.e., $\|\nabla F(v) - \nabla F(v')\|_1 \leq \beta\|v-v'\|_\infty$. Since the set of achievable values $\mathcal{V} \subseteq \left[0, \frac{1}{1-\gamma}\right]^m$, it can be verified that $\|\nabla F(v)\|_1 \leq L$ for some factor $L > 0$.

The proportional fair criterion discussed in Section 1 can be approximated by $F(v) := \sum_{i=1}^m a_i \log(\delta + v_i)$, where $\delta > 0$ is some constant introduced to circumvent the pathological case $v_i = 0$ for some $i \in [m]$. Under this criterion, $\beta = \sum_{i=1}^m a_i/\delta^2$ and $L = \sum_{i=1}^m a_i/\delta$.

When $v$ is viewed as the decision variable, at macro step $k$ with value vector $V_{1:m}^{\pi_k}(\rho)$, the ascent direction in a typical gradient ascent step is the gradient $\tilde{G}_k = \nabla_v F(V_{1:m}^{\pi_k}(\rho))$. This naturally determines the reward in the ascent direction as $\tilde{r}_k(s,a) = \langle \tilde{G}_k, r_{1:m}(s,a)\rangle$. Adapting the ARNPG framework to this specific context, we present the algorithm for solving the program (7) in Algorithm 2. We refer to it as "implicit mirror descent" because the algorithm implicitly employs mirror descent.

---

**Algorithm 2: ARNPG Implicit Mirror Descent (ARNPG-IMD)**

**Input** $\pi_0, \alpha, \eta, t_{0:K-1}, K$
**for** $k = 0, 1, \ldots, K-1$ **do**
    ⌊ Update $\pi_{k+1} \leftarrow$ InnerLoop$(\pi_k, \tilde{r}_k, \alpha, \eta, t_k)$
**Return** the policy in $\{\pi_k\}_{k=1}^K$ with the largest $F(V_{1:m}^{\pi_k}(\rho))$

---

Let $\pi^*$ be the optimal policy for (7). Based on Proposition 1, we present the following theorem which guarantees the convergence of ARNPG-IMD with appropriately selected parameters $\pi_0, \alpha, \eta, t_k$.

**Theorem 1.** *For any $K \geq 1$, take uniform policy $\pi_0$, $\alpha \geq \frac{\beta}{(1-\gamma)^3}$, $\eta = \frac{1-\gamma}{\alpha}$, and $t_k = \lceil \frac{1}{1-\gamma} \log(\frac{5LK}{\beta \log(|\mathcal{A}|)}) + 1 \rceil$. The optimality gap of ARNPG-IMD (Algorithm 2) satisfies*

$$F(V_{1:m}^{\pi^*}(\rho)) - \max_{k \in [1:K]} F(V_{1:m}^{\pi_k}(\rho)) \leq F(V_{1:m}^{\pi^*}(\rho)) - \frac{1}{K} \sum_{k=1}^{K} F(V_{1:m}^{\pi_k}(\rho)) \leq \frac{2\alpha \log(|\mathcal{A}|)}{(1-\gamma)K}. \tag{8}$$

There are a total of $K$ macro steps, and the total number of iterations is $T = \sum_{k=0}^{K-1} t_k = \Theta(\frac{K}{1-\gamma} \log(K))$. The following corollary provides the convergence rate in terms of $T$.

**Corollary 1.** *Under the same conditions as in Theorem 1, the ARNPG-IMD algorithm satisfies* $F(V_{1:m}^{\pi^*}(\rho)) - \frac{1}{K} \sum_{k=1}^{K} F(V_{1:m}^{\pi_k}(\rho)) = O\left( \frac{\beta \log(T)}{(1-\gamma)^5 T} \right)$.

*Remark.* In the absence of knowledge of $K$, we can select time-varying numbers of InnerLoop iterations, such as $t_k = \Theta(\log(k))$, and ARNPG-IMD will still have the same $\tilde{O}(1/T)$ convergence.

### 4.2 Constrained Markov decision process

Another way of trading off the objectives is to optimize one while setting hard constraints on the others. This can be formulated as the following constrained MDP (CMDP) problem:

$$\max_{\theta} V_1^{\pi_\theta}(\rho), \quad \text{s.t. } V_i^{\pi_\theta}(\rho) \geq b_i, \; \forall i \in [2:m], \tag{9}$$

where $b_{2:m} \in [0, \frac{1}{1-\gamma}]^{m-1}$. Let $\pi^* = \pi_{\theta^*}$ be the optimal policy of the CMDP problem in (9).

Define the Lagrangian of the CMDP problem as $\mathcal{L}(\pi_\theta, \lambda) = V_1^{\pi_\theta}(\rho) + \sum_{i=2}^{m} \lambda_i (V_i^{\pi_\theta}(\rho) - b_i)$, where $\lambda_i$ is the Lagrange multiplier (dual variable) corresponding to the constraint $V_i^{\pi_\theta} \geq b_i$, for each $i \in [2:m]$. The Lagrange dual function $\max_\pi \mathcal{L}(\pi, \cdot)$ is a convex function of dual variables $\lambda \geq 0$. Denote by $\lambda^*$ the optimal dual variables that minimize the Lagrange dual function. We assume $\lambda^*$ is finite, which is guaranteed by Slater's condition, i.e., there is some $\pi_\theta$ and $\xi > 0$ with $V_i^{\pi_\theta}(\rho) - b_i \geq \xi$ for any $i \in [2:m]$. Note $(\pi^*, \lambda^*)$ is a saddle point of the Lagrangian $\mathcal{L}(\pi, \lambda)$. This motivates the primal-dual approach, which iteratively performs gradient ascent for $\pi_\theta$ and gradient descent for $\lambda$. This is suitable for the CMDP setting, since for any fixed $\lambda$, the Lagrangian $\mathcal{L}(\pi, \lambda)$ corresponds to an MDP for which policy gradient can be employed.

The canonical primal-dual gradient ascent-descent method for constrained convex optimization can only guarantee $O(1/\sqrt{T})$ convergence, and consequently the primal-dual policy gradient-based approach for CMDPs [11] has the same convergence. Recently, Yu et al. [35] have proposed a primal-dual-based method with $O(1/T)$ convergence under the Euclidean setting, i.e., $B_h(x||y) = \frac{1}{2}||x-y||_2^2$. Adopting ideas from [35], we next propose the ARNPG with Extra Primal-Dual (ARNPG-EPD) algorithm (Algorithm 3). To the best of our knowledge, this new primal-dual update appears in the CMDP-related literature for the first time.

Note that $b_i - V_i^{\pi}(\rho)$ is the amount of constraint violation. There are two key ideas we adopt from [35]. The first is the design of the reward in the ascent direction

$$\tilde{r}_k(s, a) := r_1(s, a) + \sum_{i=2}^{m} (\lambda_{k,i} + \eta'(b_i - V_i^{\pi_k}(\rho))) r_i(s, a),$$

where an extra constraint violation term is added to the dual variables. The second idea is that the update of dual variables should not fall below the negative constraint violation (the first term in (10)), and it can alleviate the overshooting of dual variables. The extra constraint violation terms in $\tilde{r}_k$ and the dual update work jointly to ensure the $\tilde{O}(1/T)$ convergence.

**Theorem 2.** *For any $K \geq 1$ and $\eta' \in (0, 1]$, take uniform policy $\pi_0$, $\alpha \geq \frac{2\eta' m}{(1-\gamma)^3}$, $\eta = \frac{1-\gamma}{\alpha}$, and choose $t_k = \lceil \frac{1}{1-\gamma} \log(\frac{5L_k K}{2\eta' m \log(|\mathcal{A}|)}) + 1 \rceil$ with $L_k = 1 + \frac{\eta'(m-1)}{1-\gamma} + \sum_{i=2}^{m} \lambda_{k,i}$. The average optimality gap and the average constraint violation of ARNPG-EPD (Algorithm 3) satisfy*

$$V_1^{\pi^*}(\rho) - \frac{1}{K} \sum_{k=1}^{K} V_1^{\pi_k}(\rho) \leq \frac{3\alpha \log(|\mathcal{A}|)}{(1-\gamma)K}, \tag{11}$$

$$b_i - \frac{1}{K} \sum_{k=1}^{K} V_i^{\pi_k}(\rho) \leq \frac{1}{K} \left( \frac{2||\lambda^*||_2}{\eta'} + 3\sqrt{\frac{\alpha \log(|\mathcal{A}|)}{(1-\gamma)\eta'}} \right) \quad \forall i \in [2:m]. \tag{12}$$

**Algorithm 3: ARNPG with Extra Primal Dual (ARNPG-EPD)**

**Input** $\pi_0, \eta', \alpha, \eta, t_{0:K-1}, K$
**Initialize** $\lambda_{0,i} = \max\{\eta'(V_i^{\pi_0}(\rho) - b_i), 0\}, \ \forall i \in [2:m]$
**for** $k = 0, 1, \dots, K-1$ **do**
$\quad$ Update $\pi_{k+1} \leftarrow \text{InnerLoop}(\pi_k, \tilde{r}_k, \alpha, \eta, t_k)$
$\quad$ Update $\lambda_{k+1,i} = \max\left\{\eta'(V_i^{\pi_{k+1}}(\rho) - b_i), \lambda_{k,i} + \eta'(b_i - V_i^{\pi_{k+1}}(\rho))\right\}, \ \forall i \in [2:m]$ $\quad$ (10)
**Return:** a policy randomly chosen from $\{\pi_k\}_{k=1}^K$

---

Note that the number of micro steps $t_k$ is chosen according to the dual variables $\lambda_k$ in the previous theorem. Denote by $T := \sum_{k=0}^{K-1} t_k$ the total number of iterations.

**Corollary 2.** *Under the same conditions as in Theorem 2, the ARNPG-EPD algorithm satisfies*
$V_1^{\pi^*}(\rho) - \frac{1}{K}\sum_{k=1}^K V_1^{\pi_k}(\rho) = O(\frac{m \log(T)}{(1-\gamma)^5 T})$, *and* $b_i - \frac{1}{K}\sum_{k=1}^K V_i^{\pi_k}(\rho) = O(\frac{\sqrt{m} \log(T)}{(1-\gamma)^{2.5} T})$.

The theorem and corollary establish convergence of the average optimality gap and the average constraint violation, in the same manner as many previous works [11, 31, 12, 19] on CMDPs. However, a guarantee on the last iterate is more preferable. This drawback is inherited from the primal-dual algorithm for convex optimization, where the primal-dual algorithm with sublinear convergence can only be guaranteed on the average solution, as of our knowledge. Last iterate convergence is still an on-going open research topic.

### 4.3 Max-min trade-off criteria

Finally, we consider the max-min trade-off criterion defined as

$$\max_\theta \min_{\lambda \in \Lambda} \Phi(V_{1:m}^{\pi_\theta}(\rho), \lambda), \tag{13}$$

where $\Lambda$ is a subset of the $m$-dimensional probability simplex $\Delta([m])$. We assume $\Phi(\cdot, \lambda)$ is concave and $\Phi(v, \cdot)$ is convex. We also assume $\Phi$ is $\beta$-smooth w.r.t. the norm $\Psi(v, \lambda) = \|v\|_\infty + \|\lambda\|_1$.

The max-min criterion mentioned in Section 1 can be represented by $\Phi(v, \lambda) = \sum_{i=1}^m v_i \lambda_i / c_i$ and $\Lambda = \Delta([m])$. $\Phi$ satisfies the concave-convex assumption and is $\beta$-smooth w.r.t. the norm $\Psi$ with $\beta = O(m)$.

Denote $F(v) := \min_{\lambda \in \Lambda} \Phi(v, \lambda)$, which is concave but not necessarily smooth. Thus we cannot apply the ARNPG-IMD algorithm (Algorithm 2) due to the non-smoothness of $F$, and the subgradient-based method can only guarantee $O(1/\sqrt{T})$ convergence.

We next integrate the optimistic mirror descent ascent (OMDA) method [27] for solving minimax optimization in the ARNPG framework. Denote the gradients $\tilde{G}_k^\lambda = \nabla_\lambda \Phi(V_{1:m}^{\tilde{\pi}_k}(\rho), \tilde{\lambda}_k)$ and $\tilde{G}_k^v = \nabla_v \Phi(V_{1:m}^{\tilde{\pi}_k}(\rho), \tilde{\lambda}_k)$. It can be verified that $\|\tilde{G}_k^v\|_1 \leq L$ for some $L$ due to the smoothness of $\Phi$. OMDA performs gradient ascent along the direction $\tilde{G}_k^v$ w.r.t. the value vector, and therefore we construct the reward in the ascent direction as $\tilde{r}_k(s, a) = \langle \tilde{G}_k^v, r_{1:m}(s, a) \rangle$. OMDA performs mirror descent along direction $\tilde{G}_k^\lambda$ w.r.t. the dual vector $\lambda$. A key ingredient of OMDA is that it updates twice in each macro step. ARNPG-OMD adopts this idea and update $(\pi, \lambda)$ from the same anchor points $(\pi_k, \lambda_k)$, first with ascent direction $(\tilde{r}_k, -\tilde{G}_k^\lambda) \in \mathbb{R}^{2m}$ and then a further step with direction $(\tilde{r}_{k+1}, -\tilde{G}_{k+1}^\lambda) \in \mathbb{R}^{2m}$.

We present ARNPG-OMDA in Algorithm 4, and establish the following performance guarantees:

**Theorem 3.** *For any $K \geq 1$, take uniform policy $\pi_0$, $\eta' \leq \frac{1}{6\beta}$, $\alpha \geq \frac{6\beta}{(1-\gamma)^3}$, $\eta = \frac{1-\gamma}{\alpha}$, and $t_k = \lceil \frac{1}{1-\gamma} \log(\frac{5LK}{6\beta \log(|\mathcal{A}|)}) + 1 \rceil$. The ARNPG-OMDA algorithm (Algorithm 4) satisfies*

$$F(V_{1:m}^{\pi^*}(\rho)) - F\left(\frac{1}{K}\sum_{k=1}^K V_{1:m}^{\tilde{\pi}_k}(\rho)\right) \leq \frac{3\alpha \log(|\mathcal{A}|)}{(1-\gamma)K} + \frac{\log(m)}{\eta' K}. \tag{14}$$

**Algorithm 4: ARNPG with Optimistic Mirror Descent Ascent (ARNPG-OMDA)**

**Input** $\pi_0, \lambda_0, \eta', \alpha, \eta, t_{0:K-1}, K$
**Initialize** $\tilde{\pi}_0 = \pi_0$ and $\lambda_0, \tilde{\lambda}_0$ as uniform distribution on $[m]$
**for** $k = 0, 1, \ldots, K-1$ **do**

> Update $\tilde{\pi}_{k+1} \leftarrow$ InnerLoop$(\pi_k, \tilde{r}_k, \alpha, \eta, t_k)$, $\tilde{\lambda}_{k+1} \leftarrow \arg\min_{\lambda \in \Lambda} \{\langle \tilde{G}_k^\lambda, \lambda \rangle + \frac{D(\lambda||\lambda_k)}{\eta'}\}$
> Update $\pi_{k+1} \leftarrow$ InnerLoop$(\pi_k, \tilde{r}_{k+1}, \alpha, \eta, t_k)$, $\lambda_{k+1} \leftarrow \arg\min_{\lambda \in \Lambda} \{\langle \tilde{G}_{k+1}^\lambda, \lambda \rangle + \frac{D(\lambda||\lambda_k)}{\eta'}\}$

**Return:** a policy randomly chosen from $\{\tilde{\pi}_k\}_{k=1}^K$

Similar to the discussion after Corollary 2, Theorem 3 provides a performance guarantee on the average value vector $F(\frac{1}{K}\sum_{k=1}^K V_{1:m}^{\tilde{\pi}_k}(\rho))$, which is inherited from the OMDA methods. Denote the total number of iterations by $T := \sum_{k=0}^{K-1} 2t_k$.

**Corollary 3.** *Under the same conditions as in Theorem 3, ARNPG-OMDA satisfies* $F\left(V_{1:m}^{\pi^*}(\rho)\right) - F\left(\frac{1}{K}\sum_{k=1}^K V_{1:m}^{\pi_k}(\rho)\right) = O\left(\frac{\beta \log(T)}{(1-\gamma)^5 T}\right).$

## 5 Empirical evaluation and application

In this section, we present the experimental results on CMDP. We compare the performance of the proposed ARNPG-EPD algorithm (Algorithm 3) with two benchmarks: NPG-PD [11] and CRPO [31]. Experimental details on CMDP are postponed to Appendix A and further experiments on smooth concave scalarization and max-min trade-off are presented in Appendix B. We provide code at https://github.com/tliu1997/ARNPG-MORL.

### 5.1 Tabular CMDP with exact gradients

Recall that under softmax policy with exact gradients, Corollary 2 (Theorem 2) guarantees $\tilde{O}(1/T)$ convergence of both performance measures: average optimality gap and average constraint violation. We compare the proposed ARNPG-EPD with the benchmarks NPG-PD and CRPO under both performance measures on a randomly generated CMDP with a single constraint, which are illustrated in Figure 1. The horizontal axis is the total number of iterations, i.e., including the micro steps in InnerLoop of ARNPG-EPD.

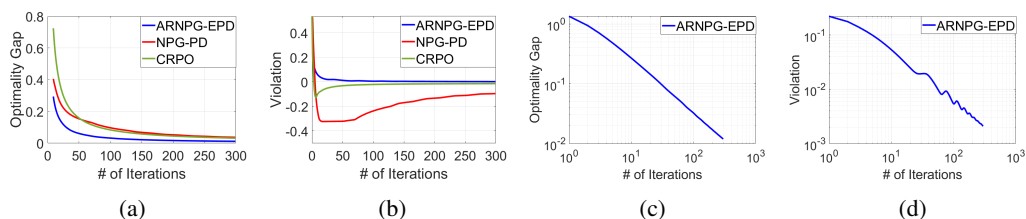

Figure 1: The average optimality gap and the average constraint violation versus the total number of iterations, for ARNPG-EPD, NPG-PD, and CRPO on a randomly generated CMDP.

Figures 1(a) and 1(b) show that both the average optimality gap and the average constraint violation of the ARNPG-EPD algorithm converge faster than those of NPG-PD. Since the CRPO focuses on the violated constraint, the policy becomes feasible quickly, though at the cost of an initially slower convergence for the optimality gap. As illustrated in Figures 1(c) and 1(d), the slopes of both the optimality gap and the constraint violation of the ARNPG-EPD algorithm in the log-log plots are approximately between -0.9 and -1, indicating a converge rate of $\tilde{O}(1/T)$.

### 5.2 Sample-based tabular CMDP

We next consider the same tabular CMDP described in Section 5.1 without exact policy gradients. Instead, policy gradients are estimated by samples from a generative model that can generate

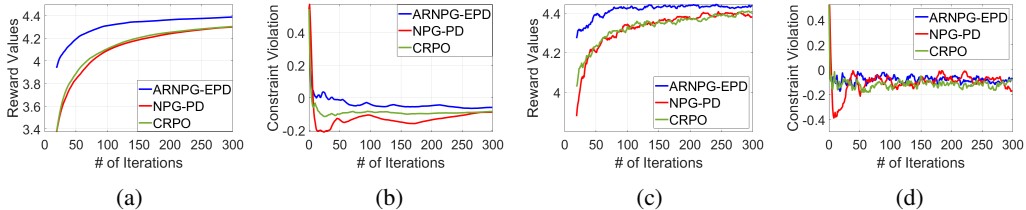

Figure 2: The reward values and the constraint violation with respect to the total number of iterations, for sample-based ARNPG-EPD, NPG-PD, and CRPO on a randomly generated CMDP.

independent trajectories starting from any state and action pair. The assumption of such a generative model is common [17, 11, 31].

The performances of CRPO, NPG-PD, and ARNPG-EPD in the sample-based scenario are shown in Figure 2. Figures 2(a) and 2(b) display the averaged performance, while Figures 2(c) and 2(d) display the performance of the current iterate (a.k.a. last-iterate in optimization literature). It shows that in this sample-based scenario, ARNPG-EPD achieves higher reward values with faster convergence, while all three algorithms satisfy the constraint after a few iterations.

### 5.3   Acrobot-v1

To demonstrate the efficacy of ARNPG-EPD on complex tasks, we have conducted experiments on the Acrobot-v1 environment from OpenAI Gym [9]. We follow the same experiment setup in [31], where there is a reward value to maximize, and two cost values to be constrained below some thresholds. The superior performance of ARNPG-EPD is shown in Figure 3.

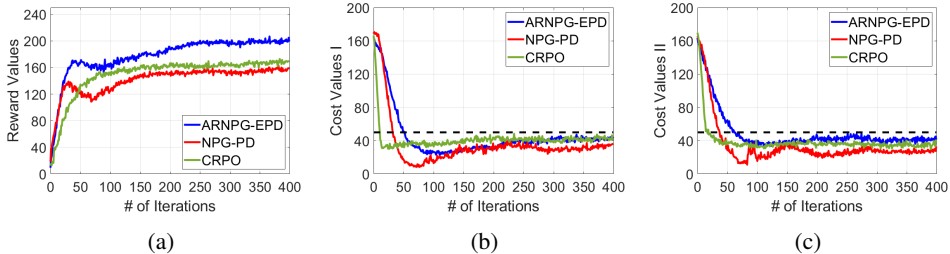

Figure 3: Last-iterate performance for sample-based ARNPG-EPD, NPG-PD, CRPO averaged over 10 random seeds. The black dashed lines in (b) and (c) represent given thresholds.

Figure 3(a) shows that ARNPG-EPD achieves a higher reward value compared to NPG-PD and CRPO, while Figures 3(b) and 3(c) demonstrate that the cost values of all three algorithms are below the thresholds after a few initial iterations. We believe the superiority is due to the new primal-dual design inspired by [34] (discussed in Section 4.2) and the flexibility of choosing $t_k$ in the InnerLoop in the framework. More experiments with different $t_k$ are presented in Appendix A.

## 6   Conclusion and future works

We propose an ARNPG framework to systematically integrate well-performing first-order methods into the design of policy gradient-based algorithms for multi-objective MDPs. The designed algorithms achieve a global $\tilde{O}(1/T)$ convergence rate under the softmax parameterization with exact gradients, and empirically have satisfactory performance beyond tabular and exact gradient settings. We believe that ARNPG has potential applications in other scenarios, since the general and flexible framework allows integration with more advanced first-order methods, currently and in the future.

A natural future direction is to extend the theoretical results to more general settings such as function approximation and sample-based scenarios. Viewing ARNPG as a heuristic, the anchor-changing ideas can also be applied to policy optimization for multi-agent RL and meta RL.

# 7 Acknowledgement

P. R. Kumar's work is based upon work partially supported by the US Army Contracting Command under W911NF-22-1-0151, US Office of Naval Research under N00014-21-1-2385, 4/21-22 DARES: Army Research Office W911NF-21-20064, US National Science Foundation under CMMI-2038625. The views expressed herein and conclusions contained in this document are those of the authors and should not be interpreted as representing the views or official policies, either expressed or implied, of the U.S. Army Contracting Command, ONR, ARO, NSF, or the United States Government. The U.S. Government is authorized to reproduce and distribute reprints for Government purposes notwithstanding any copyright notation herein.

Dileep Kalathil gratefully acknowledges funding from the U.S. National Science Foundation (NSF) grants NSF-CRII-CPS-1850206 and NSF-CAREER-EPCN-2045783.

We thank Dongsheng Ding and Tengyu Xu for generously sharing their code in [11, 31] as baselines.

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
