# OpenReview forum: "Anchor-Changing Regularized Natural Policy Gradient for Multi-Objective Reinforcement Learning"
_NeurIPS.cc/2022/Conference — NeurIPS 2022 Accept_

### Official Review · Reviewer_yFP8 · 2022-06-24

**Rating:** 5
**Confidence:** 2
**Soundness:** 3 good
**Presentation:** 3 good
**Contribution:** 3 good

**Summary:**

This paper proposed an Anchor-changing Regularized Natural Policy Gradient (ARNPG) framework, which can systematically incorporate ideas from well-performing first-order methods into the design of policy optimization algorithms for multi-objective MDP problems. Theoretically, the designed algorithms based on the ARNPG framework achieve $O(\frac{1}{T})$ global convergence with exact gradients.  A few experiments demonstrate the effectiveness of the algorithm.

**Questions:**

 There are no obvious problems. I am only concerned about whether it can be applied to more complex tasks, such as robotics control or video game, etc.

**Limitations:**

The limitations of the algorithm are rarely discussed in this paper. As far as I'm concerned, the algorithm presented in this paper is only proved to be effective in a very simple human-designed environment, and there are no relevant experiments for more practical standard tasks, such as video games.

**Strengths And Weaknesses:**

## Strengths
1. Adequate theoretical proof.
2. Well-established experimental evidence (though simple).
3. The motivation is clear.

## Weaknesses
1. Except for the tabular-based experiment, the effectiveness of ARNPG is verified in only one task (i.e., Acrobot-v1). We still need more practical examples to verify the value of the algorithm in more scenarios. I am concerned that one example alone is not enough to demonstrate the effectiveness of the algorithm.

---

> ### Author Response · Authors · 2022-08-02
> **Response to Reviewer yFP8**
>
> **Q1.** *"There are no obvious problems. I am only concerned about whether it can be applied to more complex tasks, such as robotics control or video game, etc. ...."*
>
>
> **Response:** Thank you very much for your comments and suggestions. We have now evaluated our algorithm on Hopper-v3, a more complex robotics control task, with a constraint of moving speed [R5]. Hopper-v3 is implemented via the OpenAI Gym based on the MuJoCo physical simulators [R6], where both the state space and the action space are continuous. We chose the current SOTA of this task, namely the FOCOPS (First Order Constrained Optimization in Policy Space) algorithm [R5], for comparison with our approach. Since the policy update of the FOCOPS algorithm is based on the PPO (Proximal Policy Optimization) algorithm, for a fair comparison, we also revise our algorithm to a corresponding version called ``ARPPO-EPD'', where the NPG update is replaced by PPO. Additionally, we also compare performance with other baselines NPG-PD [11] and CRPO [31]. We have now included the additional simulation in our revised manuscript, see Appendix A.3 in the revision. We are happy to report that our ARPPO-EPD algorithm performs better than other baseline algorithms, including the SOTA approach FOCOPS, even in this challenging robotic control environment.
>
>
> As the reviewer observes, the main objective of our work is to contribute to the fundamental theory of multi-objective RL and safe RL by developing a new framework to show an $\tilde{O}(1/T)$ convergence rate for several challenging multi-objective settings without strong assumptions. We really appreciate the reviewer's suggestions on more challenging environments and hope the newly added experiments would also attract attention from practical perspectives.
>
>
> [R5] Yiming Zhang, Quan Vuong, and Keith Ross. First order constrained optimization in policy space. Advances in Neural Information Processing Systems, 33, 2020.
>
>
> [R6] Emanuel Todorov, Tom Erez, and Yuval Tassa. Mujoco: A physics engine for model-based control. In 2012 IEEE/RSJ International Conference on Intelligent Robots and Systems, pages 5026–5033, 2012.

---

### Official Review · Reviewer_kSw7 · 2022-07-08

**Rating:** 7
**Confidence:** 1
**Soundness:** 3 good
**Presentation:** 4 excellent
**Contribution:** 3 good

**Summary:**

The authors propose Anchor-changing Regularized Natural Policy Gradient (ARNPG), a novel framework for designing policy-gradient based methods for multi-objective MDPs which builds on first-order methods for convex optimization problems. In contrast to prior work, it allows $\mathcal O(1/T)$ convergence without artificial or impractical assumptions. The authors theoretically analyze their approach and provide empirical evaluations in exact-gradient as well as sampling-based CMDP as well as on Acrobot.

**Questions:**

---------

**Limitations:**

-----------

**Strengths And Weaknesses:**

**Originality:** The authors develop a novel framework for the design of policy-gradient methods in multi-objective MDPs and demonstrate its application in several scenarios (proportional fairness, constrained MDP, max-min trade-off). Building on first-order convex optimization methods, they claim that their method achieves $\mathcal O(1/T)$ global convergence w.r.t. optimality gap and constraint violation, which related work does not achieve without further assumptions. They clearly work out their contributions in their Introduction and Related Work section.

**Quality:** The problem setting as well as the proposed ARNPG approach and its theoretical foundations are presented clearly and in a technically sound manner. The authors thoroughly analyze ARNPG in various different scenarios and make sure to clearly state relevant assumptions for their proofs. Further, they provide experimental evidence for the effectiveness of their approach. However, I am not familiar enough with the topic to judge the soundness of the proofs and experiments themselves.

**Clarity:** The paper is written very clearly. The authors structure their work well and do a good job in guiding the reader through their paper. I cannot judge the presentation and clarity of the derivations and proofs.

**Significance:** I believe that powerful policy-gradient methods for multi-objective MDPs and, in particular, a thorough and sound theoretical and experimental analysis as presented in the paper is important and therefore I am convinced the submission is interesting for the community. However, I am not familiar enough with the state of the art to fully judge the significance of this work.

In summary, I think this work is an interesting contribution and presented in a clear and well-written manner. Therefore, I recommend acceptance. However, this recommendation is with low confidence, as I am not very familiar with the topic. Therefore, I might adjust my rating after the discussion period.

---

> ### Author Response · Authors · 2022-08-02
> **Response to Reviewer kSw7**
>
> **Q1.** *"In summary, I think this work is an interesting contribution and presented in a clear and well-written manner. Therefore, I recommend acceptance..."*
>
>
> **Response:** Thank you very much for your careful reading of our paper and the appreciation of our work.

---

### Official Review · Reviewer_w1Sp · 2022-07-14

**Rating:** 7
**Confidence:** 3
**Soundness:** 3 good
**Presentation:** 3 good
**Contribution:** 3 good

**Summary:**

In this paper, the authors studied policy optimization in multi-objective Markov decision processes. Three specific optimization scenarios are considered: smooth concave scalarization, objective constraints, and max-min tradeoffs. The main technical contribution is an anchor-changing regularized natural policy gradient (ARNPG) framework. Theoretical analyses for the three scenarios are provided under the ARNPG framework. Some empirical experiments corroborate theoretical results.

**Questions:**

1. Line 122: equation (3), I believe it should be $v$ instead of $v_k$ in the inner product?

2. Line 639: It would improve the readability of the proof to include some more details on how equation (20) comes about, rather than simply pointing to a reference.

3. I am not entirely clear on what the “exact gradients” mean in Section 5.1. Can the authors provide some clarifications?


**Limitations:**

The authors had an adequate discussion on the limitation of the current work.


**Strengths And Weaknesses:**

Originality: This work is quite original. The authors make this connection between mirror ascent and multi-objective policy optimization. Proposition 1, which establishes an analogous inequality to classical first-order methods, is technically novel.

The theoretical applications of Proposition 1 (Section 4) are natural and instructive. The three optimization scenarios encompass a wide range of decision-making in multi-objective MDPs. In summary, this paper clearly passes the bar for originality.

Quality: I have only checked the proof of Proposition 1 (Section D in the Appendix) as it is the foundation for the subsequent analyses. The proof is technically sound and contains several interesting techniques.

Being a mainly theoretical paper, the empirical evaluation part (Section 5) provides some corroborations to the theoretical results but it is weak in terms of baselines. In Section B.1.1 Sommth concave scalarization, there are no baseline methods. Granted the specific scalarization formula (equation 16) limits baseline choices. I wonder if it is possible to apply the more common linear scalarization, which is also smooth concave, and compare with works such as the one by Yang et al. NeurIPS 2019.

Clarity: The paper is overall clear. I have some minor questions in the next section.

Significance: Theoretical analyses of policy optimization in multi-objective MDPs are highly relevant to the NeurIPS community. The novel proof techniques in this paper are likely to be significant for the future development of this topic.

---

> ### Author Response · Authors · 2022-08-02
> **Response to Reviewer w1Sp**
>
> Thank you very much for your comments and suggestions. We are encouraged by the fact that the reviewer finds our paper "original" and "technically sound and contains several interesting techniques". Please see our response below with respect to the specific comments.
>
> **Q1.** *"Granted the specific scalarization formula (equation 16) limits baseline choices. I wonder if it is possible to apply the more common linear scalarization, which is also smooth concave, and compare with works such as the one by Yang et al. NeurIPS 2019"*
>
>
> **Response:** The linear scalarization MORL setting studied in (Yang et al., 2019) is in the multi-policy setting, where the agent aims to fast adapt to the linear scalarized MORL with different linear coefficients. In this paper, we study the single-policy setting, where we aim to find a good policy for more complicated and challenging criteria, such as the constrained MDP and the minimax objective. The method given in [Yang et al., 2019] does not apply to our setting for these complicated criteria. Moreover, for the single-policy setting with linear scalarization, their setting reduces to the canonical MDP. So, our approach is not directly comparable with that of  (Yang et al., 2019).
>
>
> Our focus is on the policy gradient-based approach, and we propose the ARNPG framework with a theoretical performance guarantee. We note that the approach used in (Yang et al., 2019)  is Q-learning-based, and it does not provide any theoretical guarantee for the performance of the learned policy.
>
>
> **Q2.** *"It would improve the readability of the proof to include some more details on how equation (20) comes about ..."*
>
>
> **Response:** Thank you for the comment. We notice that Equation (20) is not utilized in our proof. To improve the readability and coherence of the proof, we decide to remove it.
>
>
> **Q3.** *"I am not entirely clear on what the “exact gradients” mean in Section 5.1. ..."*
>
>
> *Response:* The "exact gradient'' means that exact estimates of a gradient can be obtained [2]. The "exact gradient" scenarios are discussed in Lines 23-26, and the simulations in this scenario (Section 5.1) are mainly to numerically verify the theory. The "exact gradient" scenarios are common in literature (e.g., [2], [20], [10], and [11]) for studying the policy gradient-based algorithms.
>
> We thank the reviewer for pointing out the typo and we have corrected it in the revision.

---

> > ### Comment · Reviewer_w1Sp · 2022-08-09
> > **Thanks for your response**
> >
> > Thanks for answering my questions!

---

### Official Review · Reviewer_Mwr3 · 2022-07-18

**Rating:** 7
**Confidence:** 3
**Soundness:** 3 good
**Presentation:** 3 good
**Contribution:** 3 good

**Summary:**

This paper studies three instances of the multi-objective RL problem, namely smooth concave scalarization, CMDPs, and max-min trade-off. The authors proposed a new class of algorithms for these problems which updates a regularized objective which constrains the KL between the update policy and some anchor policy and uses natural gradients for policy update. The resulting algorithm achieves $\tilde{O}(1/T)$ global convergence with exact gradients.

**Questions:**

- Mathematically it seems that using natural gradients is key to achieving $\tilde{O}(1/T)$ global convergence but I wonder if you could give a little more intuition on the update rule in Equation (5), specifically if we change the gradient term to be w.r.t. the unregularized objective $V$ instead of the regularized $\tilde{V}$, this would be first-order equivalent to optimizing the KL-regularized objective in Equation (4) (first-order equivalence of TRPO and NPG) so I'm curious what is the intuition with taking the gradient of the regularized objective here?
- Following up on the previous question, what would happen if we were to remove the Fisher information term altogether and do the InnerLoop as a standard policy gradient update w.r.t. the regularized objective.
- Another question I have is related to the sample efficiency of the algorithm from a practical perspective, since both the expected KL and Fisher information matrix are w.r.t. $\pi_{\theta}$ and not the anchor policy $\pi_{\theta_k}$, this means that if we were to use a sample-based version of your proposed algorithm, we would need to collect new samples from $\pi_{\theta}$ every time we perform a policy update in the InnerLoop, wouldn't this be extremely sample inefficient even for relatively small $t_k$?
- (Minor) On line 147, is the tilde on top of the $V$ on the RHS of the equality a typo?
- (Minor) "Lagrange" is mis-spelt as "Langrange" in several places.

**Limitations:**

Yes, the authors discussed future extensions to the function approximation case

**Strengths And Weaknesses:**

I think this is a solid paper which extends the recent series of work on the global convergence of policy gradient methods to several important classes of multi-objective problems. One disclaimer I wish to make is that even though I am familiar with some of the related work, this paper is a little outside my own research area, but despite this I found the paper to be quite clearly written and fairly easy for me to follow. I did not go through all the details of the proofs but the overall structure and logic is quite clear. Multi-objective RL and constrained RL are gaining increasing interest within the community both from a theoretical and practical perspective, I think this paper makes an important contribution to our understanding of the problem and I therefore recommend this paper for acceptance.  Some additional comments and questions below in the section below.

---

> ### Author Response · Authors · 2022-08-02
> **Response to Reviewer Mwr3**
>
> We thank the reviewer for the interesting questions regarding the proposed ARNPG framework. We are encouraged by the fact that the reviewer finds our paper "quite clearly written and fairly easy to follow" and "makes an important contribution to our understanding of the problem". The specific response to questions is given below.
>
> **Q1.** *"...so,  I’m curious what is the intuition with taking the gradient of the regularized objective here"*
>
> **Response:** As the discussion at the beginning of Section 3 shows, the inner loop of the ARNPG is to approximately solve the optimization in Eq.(4). Since $\tilde{V}$ is the value we aim to maximize, we can approximate the optimizer iteratively via gradient-based methods. We view the objective in Eq.(4) as a KL-regularized MDP, and the intuition of Eq.(5) is thus to maximize the KL-regularized MDP via applying natural policy gradient to the regularized value $\tilde{V}$.
>
>
> **Q2.** *"... what would happen if we were to remove the Fisher information term altogether and do the InnerLoop as a standard policy gradient update w.r.t. the regularized objective?"*
>
>
> **Response:** The Fisher information term is the key insight of the "natural policy gradient". Removing the Fisher information term would degrade the update to the canonical policy gradient. There are both theoretical [2, 20, R1] and empirical studies [R2, R3, R4] showing the superior performance of the natural policy gradient compared to the policy gradient.
>
> [R1] Gen Li, Yuting Wei, Yuejie Chi, Yuantao Gu, and Yuxin Chen. Softmax policy gradient methods can take exponential time to converge. In Conference on Learning Theory, pages 3107–3110. PMLR, 2021.
>
> [R2] Jan Peters, and Stefan Schaal. Reinforcement learning of motor skills with policy gradients." Neural networks 21, no. 4 (2008): 682-697.
>
> [R3] Jan Peters, and Schaal, S., 2008. Natural actor-critic. Neurocomputing, 71(7-9), pp.1180-1190.
>
> [R4] Sham M Kakade. A natural policy gradient. Advances in neural information processing systems 14 (2001).
>
>
> **Q3.** *"... related to the sample efficiency of the algorithm from a practical perspective..."*
>
>
> **Response:** In the sample-based scenarios, the canonical policy gradient-based method may require new samples to estimate related quantities. This is especially true for on-policy algorithms such as NPG, TRPO, and their regularized versions. For example, applying the NPG algorithm to the regularized MDP in the sample-based scenario, the regularization term, and Fisher information matrix should both be estimated using on-policy data, i.e., the data sampled by the latest policy, at each iteration.
>
> The effectiveness of the proposed algorithm is verified numerically in the sample-based scenario, as shown in Section 5.2 and Section 5.3. Note that the x-axis (Figures 2 and 3) is the number of iterations, which has accounted for the number of updates in the InnerLoop, and a fixed amount of samples are acquired at each iteration. The proposed algorithm has a better performance compared to some baseline algorithms under the same number of samples/iterations in Figures 2 and 3. Thus instead of suffering an increase in sample complexity, the proposed algorithm may have smaller sample complexity.
>
>
> We thank the reviewers for pointing out the typos. We have corrected the typos in the revision.

---

> > ### Comment · Reviewer_Mwr3 · 2022-08-08
> > **Response to authors**
> >
> > Thank you so much for your detailed response! I am keeping my original evaluation, great work!

---

### Meta-Review · Area_Chair_aayZ · 2022-08-22

**Recommendation:** Accept
**Confidence:** Certain

**Metareview:**

This paper studies multi-objective RL, and in particular the settings of smooth concave scalarization, constraints, and max-min trade-off. The authors propose a framework for these problems called “anchor changing regularized natural policy gradient” (ARNPG) and show $O(1/T)$ convergence with exact gradients. The results do not require some of the assumptions used in related works such as ergodicity. The authors evaluate their approach on tabular environments, Acrobot, and (post-rebuttal) Hopper.
The paper is well-written, the results seem correct, and the work provides a nice extension of the recent results on PG convergence to the multi-objective case. This contribution warrants acceptance despite the somewhat limited empirical evaluation.


**Award:**

No

---

### Decision · Program_Chairs · 2022-09-14

Accept